# Bibliometric Framing of Research Trends Regarding Public Sector Auditing to Fight Corruption and Prevent Fraud

Diana-Sabina Branet [1] and Camelia-Daniela Hategan [2,*]

1 Doctoral School of Economics and Business Administration, West University of Timisoara, 300115 Timisoara, Romania; diana.branet81@e-uvt.ro
2 Department of Accounting and Audit, ECREB—East European Center for Research in Economics and Business, Faculty of Economics and Business Administration, West University of Timisoara, 300115 Timisoara, Romania
* Correspondence: camelia.hategan@e-uvt.ro

**Abstract:** Combating corruption is an important objective of the United Nations Sustainable Development Group, with the aim of helping public institutions to act in the interest of citizens. To ensure this objective is met, the spending of public money is controlled by the supreme audit institutions of each country. The objective of this paper is to identify trends in and approaches to the field of auditing in the public sector to combat corruption and prevent fraud. To achieve the proposed objective, a bibliometric analysis of papers published in the journals indexed in Web of Science Clarivate Analytics for the period 2003–2022 was carried out; selection criteria was based on instances of the keywords "public audit fraud", "supreme institution", and "fraud" appearing in a sample of 528 articles. The results showed that there was a research interest in this field, with the trend being more pronounced since 2017. The main topics addressed were those related to the performance audit and the fight against corruption, and the most relevant studies were conducted on samples from Nordic European countries. Thus, it is confirmed that the external audit in public sector is an important factor in combating the phenomenon of corruption in the public sector, both by detecting fraud and by offering recommendations aimed at making the activity of this sector more efficient.

**Keywords:** audit; public sector; corruption; fraud; prevention; institutions; bibliometric analysis

## 1. Introduction

Preventing and reducing fraud and fighting corruption is a focus of the United Nations Sustainable Development Group, outlined in Goal 16: Peace, justice, and strong institutions (United Nations 2015). This topic is intensely debated in academic and professional environments (Alamoush et al. 2021; Othman and Ameer 2023).

Dye and Stapenhurst (1998) are of the opinion that audit institutions, which are the watchdogs of a country's public financial management, must play an essential role in efforts to combat corruption. The supreme institution of each country's audit plays the most important role, through the information it provides to entities, governments, and society in general, on how to manage public resources (Mokhomole 2023). Supreme audit institutions are an important pillar of governance and the management of government resources, especially when it comes to the control of corruption (Lassou et al. 2021).

Fraudulent use of public budgets is a problem that also affects the budget of the European Union, so that public budgets are widowed by the full transfer of the due sums, on the one hand; on the other hand, the subsidies granted through community programs are not always directed towards the intended purpose for which they were granted, and, implicitly, they are not in line with the interests of the European Union (Ghinea 2012).

Research indicates that audit institutions can detect misconduct and violations in financial and public expenditure reporting and make recommendations so that management can make appropriate decisions to remedy these problems (Liu and Lin 2012). Simunic (2014)

showed that financial reporting, auditing, and other institutional procedures reduce the costs of public sector investments, as they categorically contribute to reducing fraudulent use of the state's financial resources. In-depth knowledge of corruption processes and what facilitates them is hampered by the hidden nature of the phenomenon, a fact confirmed also by Lino et al. (2022).

Therefore, the objective of this paper is to identify trends in and approaches to auditing in the public sector, with the aim of combating corruption and preventing fraudulent use of public funds. Therefore, the research objective is based on the question: Does the audit in the public sector make a substantial contribution to combating corruption and preventing fraud?

Methodologically, this research is based on the bibliometric analysis of research published on this subject, using a sample of 528 papers published between 2003–2022 in journals indexed in the Web of Science–Clarivate Analytics (WoS), using VOSviewer software 1.6.20.

This paper contains an analysis of the available data on public sector auditing from the perspective of research on its contribution to the fight against corruption and the importance of supervising the supreme audit institutions' spending of public money. The paper can be a relevant source for researchers in the audit field and public sector management to understand the need for and importance of the independence of auditors' missions in the public sector. The paper contributes to the existing research by synthesizing the information regarding public sector performance and its monitoring through the audits performed.

This paper is structured as follows: in the following section, the literature on public sector auditing will be reviewed; in Section 3, the research methodology will be explained; and, in Section 4, the results obtained will be presented. The paper ends with conclusions, research limitations, and future directions.

## 2. Literature Review

The existing literature on public sector auditing and anti-corruption and anti-fraud regulations provides insight into the nature of this pervasive problem. The topic has been researched in different dimensions, but the aim of this paper is to present it from the perspective of the respective auditors of the supreme audit institutions.

Supreme Audit Institutions (SAIs) play important roles within the institutional mechanisms of the democratic state. They are given a high degree of independence to ensure, firstly, public accountability, probity, and legality of public expenditure; and, secondly, economy, efficiency, and effectiveness (Pollitt and Summa 1997).

International pressures on Supreme Audit Institutions (SAIs) to fight corruption are increasing. SAIs have a notable impact on reducing corruption, especially when they are entrusted with more sanctioning powers and their audits are carried out independently and professionally. SAIs often enjoy higher levels of citizen trust, compared to other government institutions (Tara et al. 2016) that contribute to anti-corruption practices, in two main ways: deterrence and detection (Dye 2007).

The active participation of SAIs that have financial, organizational, operational, and functional independence, in addition to audit mandates that are guaranteed by constitutions and laws, is essential for deterring corruption by promoting accountability, openness, and good governance in public financial management. In addition, the SAI is likely to detect some suspected cases of corruption, even though most do not have the power to investigate those cases (Reichborn-Kjennerud 2013).

Building strong institutions is a central development challenge and is key to controlling corruption. Among public institutions, Supreme Audit Institutions (SAIs) play a key role as they contribute to promoting sound financial management and thus accountable and transparent governance (Dye and Stapenhurst 1998).

Independence is a key feature of SAIs that can be crucial; to be effective in the fight against corruption, they must first be fully independent from national governments, although attempts are sometimes made to curtail this independence.

Gustavson and Sundström (2018) considered that, as a result of different legal and institutional frameworks, the mandate and exact nature of SAIs' audit duties varies from country to country, and there is also great diversity in terms of structure, professionalism, size, independence of resources, and transparency. Auditing in the public sector is a vital activity in democratic states, underpinning the relationship between the government and the governed, the executive and the legislature and various parts of government (Ferry et al. 2022).

External public audit activity is imperative in ensuring public sector accountability (Cordery and Hay 2019). Supreme Audit Institutions can have direct and indirect contributions, respectively, to the fight against corruption; through detection or deterrence, depending on their organization and strategy, as well as the political culture and the perception of corruption (Kayrak 2008).

In the past, Supreme Audit Institutions rarely made their work public, but lately they have become more concerned with communication. A communication policy completes their cycle of responsibility, justifies their existence, and is an essential component of their independence and effectiveness; it also brings measures that evaluate the impact of their activity (González et al. 2008).

Effective communication with interested parties, achieved by increasing the degree of transparency of information (Hategan et al. 2015), is key for organizations, both public and private ones (Hategan 2021). Therefore, it can be appreciated that the role of Supreme Audit Institutions in combating corruption is the communication of planning, as well as audit strategies; facilitating the exchange of information; and the organization of programs as well as training courses.

To highlight the importance and responsibilities of the Supreme Audit Institutions, comparative studies were carried out, from which it emerged that there is diversity in the organization, capacities, and scope of the SAI in each country (Ferry et al. 2023b) and that the performance audit is being applied with increasing frequency (Ferry et al. 2023a).

Qualitative research was also carried out that systematically studied and revised the published literature on the value of auditing in the public sector (Hay and Cordery 2018), its contribution to the fight against corruption (Assakaf et al. 2018), and the role of the performance audit (Rana et al. 2022). In recent years, an easier tool was applied, i.e., bibliometric analysis, and a close connection between auditing and fraud prevention was found (Filatova et al. 2023), but research using this approach is still limited. Thus, our study aims to fill the gap by highlighting the connection between auditing in the public sector and the phenomena of corruption and fraud. The bibliometric analysis of the articles will identify existing research directions, research gaps, and potential opportunities for future research.

## 3. Materials and Methods

The bibliometric analysis was carried out to determine the direction and approaches in the field of public sector auditing and fraud prevention, and the researchers most concerned with the topic under study. The data processing was carried out with the help of VOSviewer software 1.6.20, with the aim of identifying the relationships between the most common keywords in the literature, as well as those between the authors, their research, and related citations. The data used were downloaded from Web of Science (WoS)—Clarivate Analytics on 30 June 2023; the papers were selected from a time period spanning 20 years, specifically, 2003–2022. To identify the papers that addressed the topic of auditing in the public sector and the prevention of fraud, expressions that contained the keywords "public audit fraud", "supreme institution", and "fraud", in addition to the topic, were selected. The database initially highlighted 990 papers belonging to all document categories. The documents were filtered according to the following criteria: the document type (article) and the WoS category (with minimum 25 record count). In the analyzed period 2003–2022, after applying the selection criteria, 528 papers were identified (Figure 1).

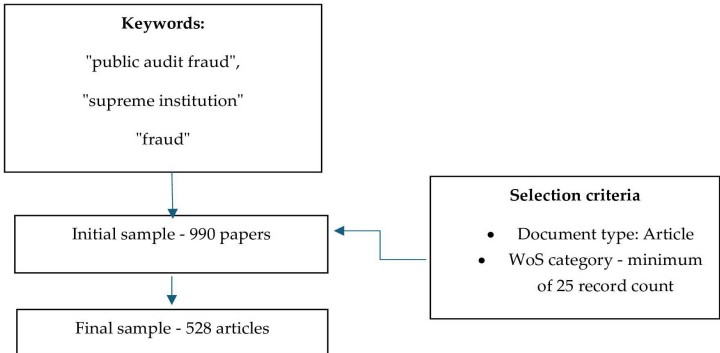

**Figure 1.** Sample selection. Source: Own processing based on data from WoS.

To improve word selection, similar expressions or abbreviations were replaced and meaningless words were removed. The final database was subsequently processed with the help of VOSviewer software 1.6.20, which resulted in the analysis of keywords and citations based on the distribution of papers by the individual authors and the countries, carried out after their affiliation.

## 4. Results

Figure 2 shows the distribution by year of the articles published on the studied topic.

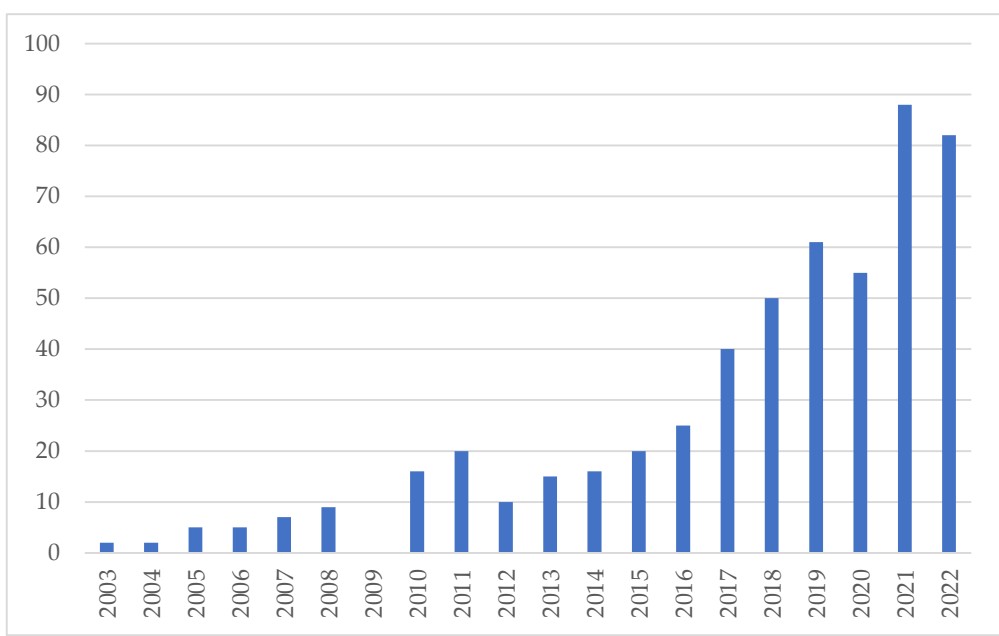

**Figure 2.** Annual evolution of published articles. Source: Own processing based on data from WoS.

From Figure 1 it can be observed that, in the first years of the studied period, the number of articles was reduced to below 10 articles per year. Starting with 2008, a slightly increasing trend is observed until 2011, after which a slight decrease is noted; but, starting with 2017, the number of articles almost doubles compared to the previous year. The number of published papers reaches over 80 articles in 2022, a possible explanation being the fact that the subject came to the attention of several authors, a fact that confirms the importance of the studied topic, a situation also confirmed by the research of Maggiorani (2022).

For a detailed analysis, the papers were grouped according to the domains existing in WoS (Table 1).

**Table 1.** Web of Science Categories.

| Web of Science Categories | Record Count | % of 528 |
|---|---|---|
| Business Finance | 226 | 42.80 |
| Public Administration | 157 | 29.73 |
| Management | 111 | 21.02 |
| Economics | 80 | 15.15 |
| Business | 45 | 8.52 |
| Political Science | 26 | 4.924 |

Source: Own processing based on data from WoS.

Based on the settings presented in the methodology section, six categories were highlighted; thus, from Table 1, it follows that the main research area of Business Economics had the largest weight, followed by Public Administration and Political Science. Considering that some journals were indexed in several fields, it is not possible to total the number of papers by field.

Analysis by keywords shows the way in which the most common keywords were found together in the analyzed papers. A total of 2218 keywords were highlighted in the VOSviewer software 1.6.20. From the total number of words, those that registered at least five interactions were selected in the analysis, resulting in a selection of 158 words, from which 20 words were eliminated, represented by the irrelevant ones and those with similar expressions (for example, the union of the terms expressed in singular or plural). Therefore, 138 words were kept in the final analysis, set in the bibliometric software to a minimum of 10 words per cluster, for resolution one (Figure 3).

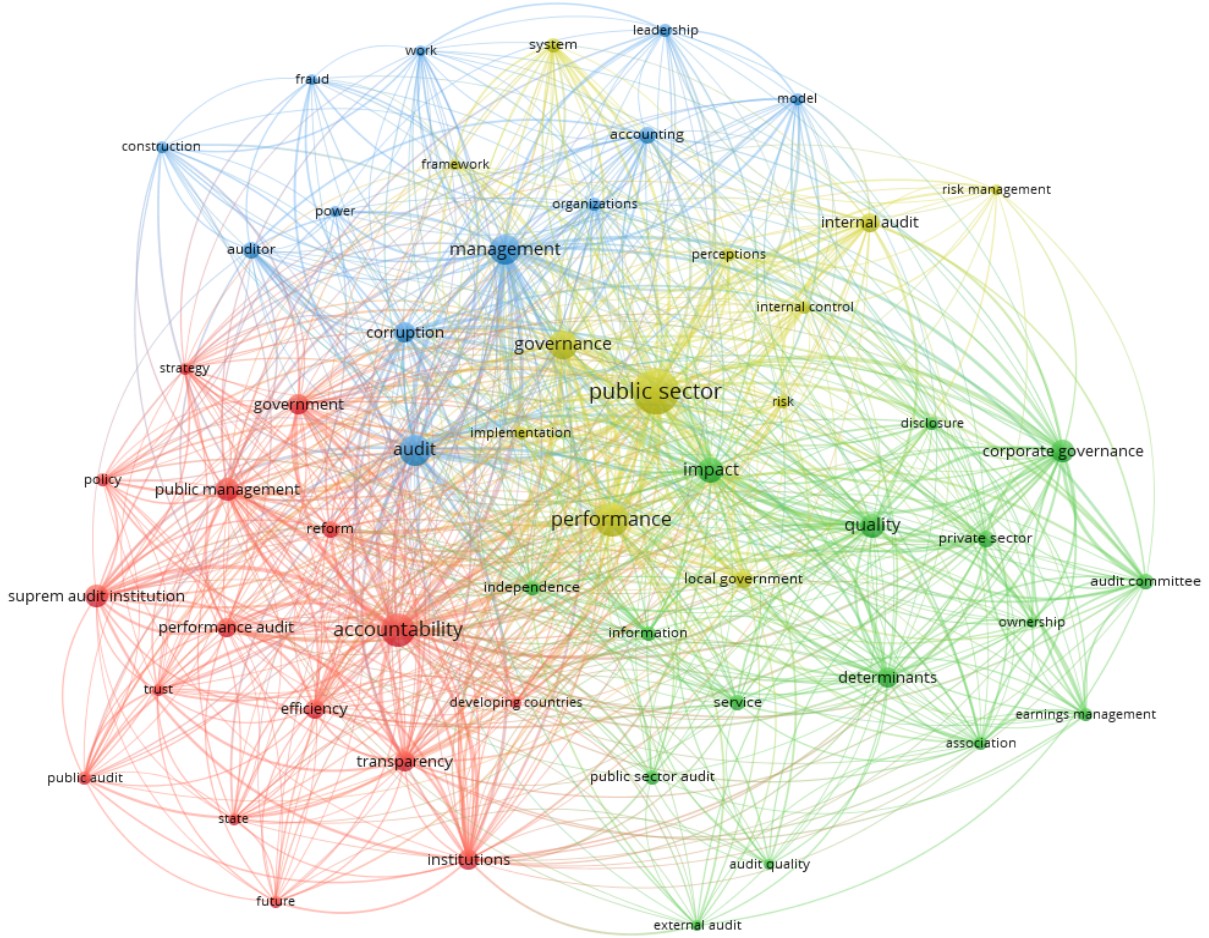

**Figure 3.** Bibliometric map of keywords. Source: Own processing based on data from WoS.

As shown in Figure 3, the keywords were grouped into four clusters. Table 2 shows the clusters of words that interacted with one another, in descending order according to the number of interactions. Cluster 1 comprises the main keywords, referring to "public sector" and "performance" that made connections with the other clusters; respectively, Cluster 2, with the key element "responsibility" (accountability); Cluster 3, highlighted by the keywords "audit" and "management"; and Cluster 4, including the words "impact" and "quality", which had the most links within the cluster.

**Table 2.** Distribution of keywords in clusters.

| Cluster 1 (Yellow) | No. | Cluster 2 (Red) | No. | Cluster 3 (Blue) | No. | Cluster 4 (Green) | No. |
|---|---|---|---|---|---|---|---|
| Public sector | 152 | Accountability | 84 | Audit | 72 | Impact | 47 |
| Performance | 83 | Supreme audit institution | 41 | Management | 71 | Quality | 46 |
| Governance | 61 | Public management | 38 | Corruption | 34 | Corporate governance | 40 |
| Internal audit | 27 | Institutions | 34 | Auditor | 22 | Determinants | 32 |
| Local government | 24 | Government | 32 | Accounting | 21 | Private sector | 21 |
| System | 16 | Transparency | 31 | Leadership | 13 | Audit committee | 18 |
| Perceptions | 15 | Efficiency | 30 | Organizations | 13 | Service | 18 |
| Risk | 14 | Performance audit | 30 | Construction | 12 | Independence | 17 |
| Implementation | 13 | Reform | 21 | Model | 12 | Information | 17 |
| Internal control | 13 | Policy | 15 | Fraud | 10 | Public sector audit | 16 |

Source: Own processing based on data from WoS.

Table 2 shows that Cluster 1 grouped the words with the most interactions, starting from the central keyword "public sector" with 152 links, which was followed by other words such as "performance" and "governance", words that confirm that the research topic was a previously studied one. The second cluster refers to "accountability", which also contains elements derived from the connection with the supreme institutions of auditing: "transparency", "policies", and "reforms". The third cluster related to audit also contains links to the words "corruption" and "fraud", which confirms the premises on which this research started. Fewer interactions were observed in Cluster 4, which has, as its central elements, "impact", "quality", and "independence", and also highlights aspects from the field of external audit of private entities.

The distribution of papers by country, after the affiliation of the authors, was achieved by selecting a minimum of three documents/country and a minimum of three citations/country, with a minimum of four elements in the minimum size of a cluster. The results obtained showed that the authors come from 39 countries (Figure 4).

From Figure 4, the authors' countries of origin were grouped into four clusters, whose detailed presentation can be found in Table 3.

Table 3 shows that the most frequent authors were grouped in Cluster 3, represented by the United Kingdom of Great Britain, consisting of England (70 papers) alongside Wales (11 papers). In the same cluster are other countries with English-speaking population (South Africa, New Zealand), a situation that can be explained by the ease with which research teams can be formed or the easier understanding of written articles. The next cluster (2) is represented by Australia, along with the USA and Canada, which are other predominantly English-speaking countries. Authors mainly from European Union or European countries were grouped in Cluster 1, a cluster in which most of the selected papers were by authors from Spain, a country followed by Sweden and Italy. Romania was also included in this cluster, with seven papers published by Romanian authors. Fewer countries were included in Cluster 4, of which Brazil had the most observations, followed by papers by authors from Ghana and other European countries (Germany, Ukraine).

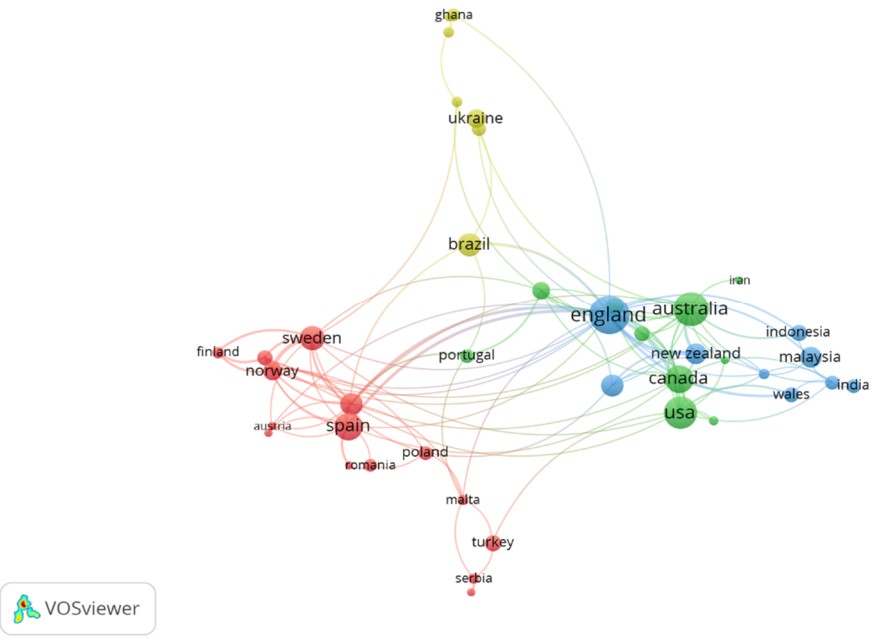

**Figure 4.** Bibliometric map of reference countries. Source: Own processing based on data from WoS.

**Table 3.** Distribution of authors by country.

| Cluster 1 (Red) | No. | Cluster 2 (Green) | No. | Cluster 3 (Blue) | No. | Cluster 4 (Yellow) | No. |
|---|---|---|---|---|---|---|---|
| Spain | 36 | Australia | 54 | England | 70 | Brazil | 25 |
| Sweden | 28 | USA | 50 | Indonesia | 13 | Cyprus | 3 |
| Italy | 16 | Canada | 36 | Malaysia | 21 | France | 5 |
| Norway | 13 | Netherlands | 14 | New Zealand | 21 | Germany | 9 |
| Turkey | 11 | Belgium | 11 | China | 10 | Ghana | 8 |
| Denmark | 10 | Portugal | 9 | South Africa | 23 | Greece | 5 |
| Poland | 8 | Saudi Arabia | 4 | Arab emirates | 5 | Ukraine | 9 |
| Romania | 7 | Iran | 3 | Wales | 11 | | |
| Finland | 5 | Singapore | 3 | | | | |
| Serbia | 5 | | | | | | |

Source: Own processing based on data from WoS.

Therefore, the topic has been studied by authors from all continents, from Europe in all clusters, from Asia in clusters 2 and 3, from Africa in clusters 3 and 4, from Australia in clusters 2 and 3, from North America in Cluster 2 and from South America in Cluster 4.

The distribution of papers among the universities to which the authors are affiliated was analyzed by selecting a minimum of two documents/country, with a minimum of two citations/country and a minimum of four elements included in the cluster, with the results being presented in Figure 5.

Figure 5 shows that the universities with which the authors were affiliated were grouped into five clusters, and their detailed presentation is made in Table 4.

Table 4 shows that the universities were relatively equally grouped in the five clusters, so the maximum number of papers, in three clusters, was nine. from the countries that stood out in the distribution of authors by country; namely, University South Africa (Africa de South) University Zaragoza (Spain), and University Essex (England). The following affiliations, with eight papers, belonged to authors from Rmit University and University Auckland.

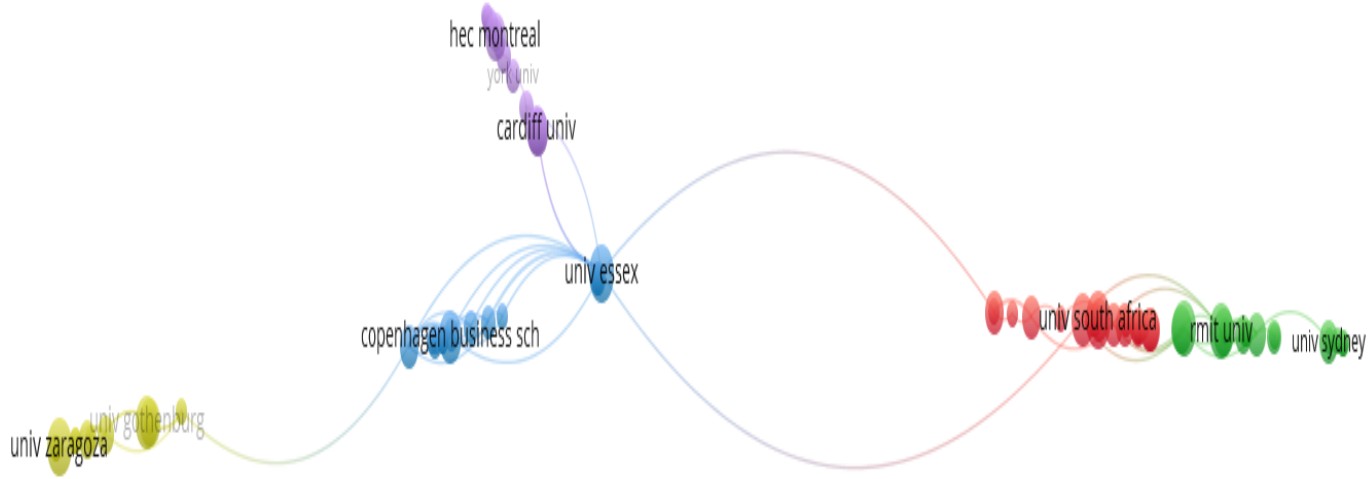

**Figure 5.** Bibliometric visualization of reference universities. Source: Own processing based on data from WoS.

**Table 4.** Distribution of authors by universities.

| Cluster 1 | Documents | Cluster 2 | Documents | Cluster 3 | Documents |
|---|---|---|---|---|---|
| University South Africa | 9 | Rmit University | 8 | University Zaragoza | 9 |
| Massey University | 7 | University Auckland | 8 | University Gothenburg | 7 |
| Tshwane University Technol | 5 | Aston University | 6 | Lund University | 6 |
| University Birmingham | 5 | University Pretoria | 6 | Kristianstad University | 4 |
| University Canberra | 5 | Macquarie University | 5 | Nord University | 4 |
| University Nottingham | 5 | University Sydney | 5 | Kozminski University | 2 |
| University Teknol Mara | 5 | University Glasgow | 4 | | |
| University Utara Malaysia | 5 | University New S Wales | 3 | | |
| **Cluster 4** | | **Cluster 5** | | | |
| University Essex | 9 | Cardiff University | 6 | | |
| Copenhagen Business Sch | 7 | Hec Montreal | 6 | | |
| Stockholm University | 5 | Rutgers State University | 6 | | |
| Fundacao Getulio Vargas | 3 | University Durham | 6 | | |
| Oslo Metropolitan University | 3 | Leiden University | 3 | | |
| Tampere University | 3 | Roskilde University | 3 | | |

Source: Own processing based on data from WoS.

Highlighting the most cited authors who researched aspects of external auditing in the public sector, combating corruption and fraud prevention was analyzed by selecting, in the database, at least two documents/country, with at least two citations/country and at least four elements included in the cluster, with the results being presented in Figure 6.

Figure 6 shows that the most cited authors were grouped into three clusters whose detailed presentation can be found in Table 5.

The analysis bibliometric showed that the most cited authors were grouped in Cluster 1 as follows: Jeppesen, with 134 citations; and Reichborn-Kjennerud, with 91 citations. The ranking continues with Johnsen and Carrington, with more than 70 accumulated citations each. In Cluster 2, the author with the most citations was Steccolini, with 40 citations. Graham stands out in Cluster 3, with 117 citations; and, in Cluster 4, the significant author was Ferry (117 citations). Ferry is the author with the most articles in the sample, but considering that most articles were published at the end of the analyzed period, he did not

accumulate many citations. Also, it can be seen that Steccolini is the author who makes the connection between all four clusters, having numerous collaborations with authors from other universities.

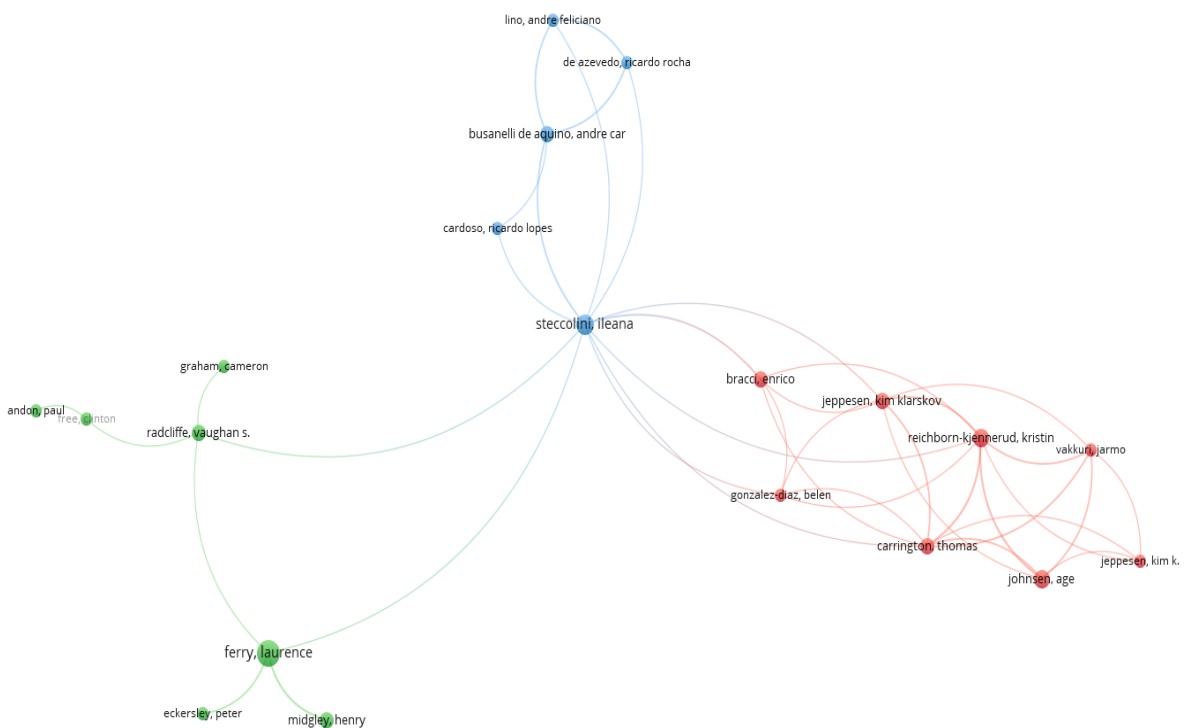

**Figure 6.** Bibliometric map of reference authors. Source: Own processing based on data from WoS.

**Table 5.** Distribution of authors by clusters.

| Cluster 1 | Documents | Citations | Avg. Citations | Cluster 2 | Documents | Citations | Avg. Citations |
|---|---|---|---|---|---|---|---|
| Jeppesen, Kk | 5 | 134 | 26.80 | Steccolini, I | 5 | 40 | 8.00 |
| Reichborn-Kjennerud, K | 4 | 91 | 22.75 | Busanelli De Aquino, Ac | 3 | 13 | 4.33 |
| Johnsen, A | 4 | 72 | 18.00 | De Azevedo, Rr | 2 | 11 | 5.50 |
| Carrington, T | 3 | 71 | 23.67 | Lino, Af | 2 | 11 | 5.50 |
| Vakkuri, J | 2 | 50 | 25.00 | Cardoso, Rl | 2 | 8 | 4.00 |
| Gonzalez-Diaz, B | 2 | 40 | 20.00 | | | | |
| Bracci, E | 3 | 32 | 10.67 | | | | |
| **Cluster 3** | | | | **Cluster 4** | | | |
| Graham, C | 2 | 117 | 58.50 | Ferry, L | **8** | **117** | 14.63 |
| Andon, P | 2 | 103 | 51.50 | Eckersley, P | 2 | 70 | 35.00 |
| Free, C | 2 | 61 | 30.50 | Zakaria, Z | 2 | 34 | 17.00 |
| Radcliffe, Vs | 3 | 20 | 6.67 | Midgley, H | 3 | 26 | 8.67 |

Source: Own processing based on data from WoS.

The articles were published in several journals whose distribution, in descending order, is presented in Table 6.

**Table 6.** Distribution of papers by publication titles.

| Publication Titles | Record Count | % of 528 |
| --- | --- | --- |
| Public Money Management | 38 | 7.20 |
| Financial Accountability Management | 32 | 6.06 |
| Accounting Auditing Accountability Journal | 19 | 3.60 |
| Journal of Public Budgeting Accounting Financial Management | 17 | 3.22 |
| Managerial Auditing Journal | 13 | 2.46 |
| Public Finance Quarterly Hungary | 10 | 1.89 |
| Southern African Journal of Accountability and Auditing Research Sajaar | 10 | 1.89 |
| Australian Accounting Review | 9 | 1.71 |
| Public Management Review | 9 | 1.71 |
| Contemporary Issues in Public Sector Accounting and Auditing | 7 | 1.14 |
| Contemporary Studies in Economic and Financial Analysis | 7 | 1.14 |
| Independent Journal of Management Production | 7 | 1.14 |
| International Review of Administrative Sciences | 7 | 1.14 |
| Public Administration | 7 | 1.14 |
| Abacus A Journal of Accounting Finance and Business Studies | 6 | 1.14 |
| Critical Perspectives on Accounting | 6 | 1.14 |
| Energy Policy | 6 | 1.14 |
| Journal Of Business Ethics | 6 | 1.14 |
| Administration & Society | 5 | 0.95 |
| Cogent Business Management | 5 | 0.95 |
| International Journal of Auditing | 5 | 0.95 |
| Lex Localis Journal of Local Self Government | 5 | 0.95 |
| Quality Access to Success | 5 | 0.95 |
| Other publications with less 5 papers | 287 | 54.35 |
| **Total** | **528** | **100.00** |

Source: Own processing based on data from WoS.

In Table 6, the journals that published at least five articles from the selected sample were included. Most papers were published in Public Money Management (38), Financial Accountability Management (32), and Accounting Auditing Accountability Journal (19). The journal with the most articles included in a sample is indexed in the Public Administration category; on the other hand, the following two journals are from the field of Business Economics, a topic being increasingly published by journals in this field, which also leads to a higher number of citations of the published articles.

More than half of the number of publications recorded fewer than five articles in the analyzed period. This fact shows that the topic studied is of wide interest, being published by a large number of journals.

To identify the most relevant articles on the subject studied in Table 7, the first 10 articles are ordered according to the number of citations accumulated since publication.

Table 7 shows that the most relevant articles were published in the second part of the period selected in the analysis, namely, in the last 10 years. Thus, it turns out that the article written by Andon (2012) accumulated the most citations (56) with an annual average of 5.1, and the article with the highest annual citations, at 9.75, was the article by Jeppesen (2019) who accumulated a considerable number of citations (39). Moreover, Jeppesen is the

author with the most citations (140) from all the papers analyzed (five), as it results from Table 6, where he was included in Cluster 1.

**Table 7.** The 10 most cited articles.

| Title | Authors | Journal | Year | Citations | Aver/Year |
|---|---|---|---|---|---|
| Accounting-related research in PPPs/PFIs: present contributions and future opportunities. | Andon, P. | Accounting, Auditing & Accountability Journal | 2012 | 56 | 5.1 |
| Political accountability and performance audit: the case of the auditor general in Norway. | Reichborn-Kjennerud, K | Public Administration | 2013 | 43 | 4.3 |
| The role of auditing in the fight against corruption | Jeppesen, K. K | The British Accounting Review | 2019 | 39 | 9.75 |
| A 'panoptical' or 'synoptical' approach to monitoring performance? Local public services in England and the widening accountability gap. | Eckersley, P., Ferry, L., & Zakaria, Z. | Critical Perspectives on Accounting | 2014 | 33 | 3.7 |
| The strategic options of supreme audit institutions: The case of four Nordic countries | Jeppesen, K. K., Carrington, T., Catasús, B., Johnsen, Å., Reichborn-Kjennerud, K., & Vakkuri, J | Financial Accountability & Management | 2017 | 27 | 4.5 |
| Public sector audit in contemporary society: A short review and introduction. | Johnsen, Å | Financial Accountability & Management | 2019 | 23 | 5.75 |
| Communication as a transparency and accountability strategy in supreme audit institutions | González-Díaz, B., García-Fernández, R., & López-Díaz, A. | Administration & Society | 2013 | 19 | 1.9 |
| Supreme audit institutions in a high-impact context: A comparative analysis of performance audit in four Nordic countries | Johnsen, Å., Reichborn-Kjennerud, K., Carrington, T., Jeppesen, K. K., Taro, K., & Vakkuri, J. | Financial Accountability & Management | 2019 | 18 | 4.5 |
| Sais work against corruption in Scandinavian, South-European and African countries: An institutional analysis | Reichborn-Kjennerud, K., González-Díaz, B., Bracci, E., Carrington, T., Hathaway, J., Jeppesen, K. K., & Steccolini, I. | The British Accounting Review | 2019 | 18 | 4.5 |
| Performance audits and supreme audit institutions' impact on public administration: The case of the office of the auditor general in Norway | Reichborn-Kjennerud, K., & Johnsen, Å. | Administration & Society | 2018 | 18 | 3.6 |

Source: Own processing based on data from WoS.

It is also observed that Reichborn-Kjennerud was co-author in most articles (five articles), followed by Jeppesen and Johnsen (four articles) and Carrington (three articles). The four authors who are co-authors of several articles were grouped in Cluster 1 in Table 6.

The journals in which the most articles were published is Financial Accountability & Management (three articles) indexed in ESCI (Business, Finance), publisher Willey, a journal that was ranked second according to the number of papers selected. The ranking is followed by The British Accounting Review (two articles) indexed in SSCI (Business, Finance), publisher Elsevier. The other five articles have been published in five different journals.

The objective of the research of the 10 articles from Table 7 will be presented in the following sentences.

Andon (2012) examined the literature on the role and effects of accounting in public–private partnerships, from which he derived several research opportunities in this field, resulting in the fact that the complexity of these partnerships must be considered, an important role falling to the performance audit in the name of the public interest.

Reichborn-Kjennerud (2013) analyzed the perceptions of civil servants in Norway regarding the usefulness of performance auditing. The results of the questionnaire showed that the performance audit was considered useful by most public administration officials, having an impact on the officials' responsibility.

Jeppesen (2019) studied how different types of auditing can contribute to the fight against corruption, concluding that, for auditing to gain a more prominent role in the fight against corruption, auditing standards must include corruption in their definition of fraud, private and public sector auditors must cooperate and share information, audit techniques should be used to detect corruption, and the auditing profession must adopt effective preventive measures such as anti-corruption certifications.

Eckersley et al. (2014) analyzed the impact of audit reforms in England by abandoning the centralized monitoring regime advocating a transparency of financial information so that citizens can request answers directly from local public institutions. The identified regulatory risk refers to the fact that citizens will not assume this role of evaluating the performance of public services, a function that could be transferred to outsourcing companies.

Jeppesen et al. (2017) investigated four strategic options for Supreme Audit Institutions regarding the performance audit; a financial audit; a portfolio strategy; and a hybrid strategy. The analysis was carried out in the case of four Nordic ISAs and showed that one ISA (Denmark) seems to have adopted a hybrid strategy, and the other three (Finland, Norway, and Sweden) chose the portfolio strategy.

Johnsen (2019) carried out a brief review of recent research on auditing in the public sector and found that there are large variations in the way audit institutions are organized, an important factor being the independence and relevance of auditors, in determining their effectiveness in fighting corruption.

González-Díaz et al. (2013) analyzed the communication methods of the Supreme Audit Institutions with the interested public and found that the communication strategy must be based on the target audience, the message sent, and the communication channels used to fight corruption and ensure transparency and accountability in government activity.

Johnsen et al. (2019) analyzed the impact of the performance audit of Supreme Audit Institutions (SAIs) on public administration in four Nordic countries (Denmark, Finland, Norway, and Sweden). The statistical regression results indicated that performance audits had a positive impact on utility, changes, improvements, and, to some extent, accountability on public administration employees that managed the performance audit made by SAI auditors. The most important factors identified were the legitimacy of the SAI, the quality of the audit and the consequences of communication in the mass media.

Reichborn-Kjennerud et al. (2019) compared seven SAIs from Scandinavian, Southern European, and African countries to better understand how these institutions perceive their role in fighting corruption. The results showed that there is no direct connection between their activity and the level of corruption in each country. National legislation that attempts to combat corruption through coercive measures has a notable influence, so the influence of INTOSAI, which must obtain increased institutional recognition worldwide, can be considered limited.

Reichborn-Kjennerud and Johnsen (2018) analyzed the tendency of audited public institutions to make changes, as a result of performance audits by Supreme Audit Institutions. The research was based on a survey of civil servants from public institutions in Norway and the results showed that government institutions tended to make changes, but these depended on instrumental, institutional, and political factors.

Previously, the most relevant papers were presented, which can be grouped into two major themes addressed, as shown in Table 8.

**Table 8.** The research topics.

| Topic | Paper Title | Autor |
|---|---|---|
| Audit performance | Accounting-related research in PPPs/PFIs: present contributions and future opportunities. | Andon, P. |
| | Political accountability and performance audit: the case of the auditor general in Norway. | Reichborn-Kjennerud, K |
| | A 'panoptical' or 'synoptical' approach to monitoring performance? Local public services in England and the widening accountability gap. | Eckersley, P., Ferry, L., & Zakaria, Z. |
| | Supreme audit institutions in a high-impact context: A comparative analysis of performance audit in four Nordic countries | Johnsen, Å., Reichborn-Kjennerud, K., Carrington, T., Jeppesen, K. K., Taro, K., & Vakkuri, J. |
| | Performance audits and supreme audit institutions' impact on public administration: The case of the office of the auditor general in Norway | Reichborn-Kjennerud, K., & Johnsen, Å. |
| Fighting corruption | The role of auditing in the fight against corruption | Jeppesen, K. K |
| | The strategic options of supreme audit institutions: The case of four Nordic countries | Jeppesen, K. K., Carrington, T., Catasús, B., Johnsen, Å., Reichborn-Kjennerud, K., & Vakkuri, J |
| | Public sector audit in contemporary society: A short review and introduction. | Johnsen, Å |
| | Communication as a transparency and accountability strategy in supreme audit institutions | González-Díaz, B., García-Fernández, R., & López-Díaz, A. |
| | Sais work against corruption in Scandinavian, South-European and African countries: An institutional analysis | Reichborn-Kjennerud, K., González-Díaz, B., Bracci, E., Carrington, T., Hathaway, J., Jeppesen, K. K., & Steccolini, I. |

Source: own processing.

Table 8 shows that the audit performance category included papers that addressed aspects of public–private partnership administration, political responsibility, performance monitoring, comparative analyses between countries, and the impact on public administration. In the category of combating corruption, topics such as the role of the audit, a review of the literature on the public sector audit, the strategic options of the SAI, institutional communication, and an international institutional analysis were addressed.

In addition to the relevant research presented, it is possible that there are other valuable studies that were not highlighted by the bibliometric analysis of the data extracted from the Web of Science.

Previous bibliometric analysis related to public sector referred to the performance audit (Marthin et al. 2021), audit quality (Maggiorani 2022), accountability and transparency (Amalia 2023), and performance management in the public sector (Roy et al. 2023).

Compared to the previous bibliometric analysis related to the public sector, the originality of our paper is provided by the association between auditing in the public sector and performance and the fight against corruption.

## 5. Conclusions

The literature review, as well as the conducted study, highlighted the fact that the external public audit is an important factor in combating the phenomenon of corruption in the public sector, both by detecting fraud and by providing recommendations aimed at making the activity of this sector more efficient. Supreme Audit Institutions contribute to combating corruption phenomena by ensuring the communication of audit planning and strategies, by facilitating the exchange of information, by organizing programs and joint training courses, or by publishing audit reports to discourage these practices.

The aim of this paper was to identify the published research on the topic of auditing in the public sector through bibliometric analysis of the links between key concepts, authors,

number of accumulated citations, and journals in which the articles were published, as well as a systematic review of the most relevant published articles.

The results of the bibliometric analysis showed that research interest in the studied topic had an increasing trend in the second part of the studied period; a possible explanation would be that of the positive evolution of the countries' economic development. Close links were confirmed between the analyzed concepts, specifically "public audit fraud", "supreme institution", and "fraud".

The main conclusion that emerges is that research in this field has a wide spectrum, both in developed countries and in less developed countries, the articles have been published in many journals, which shows a continued concern for monitoring the public sector. This topic can be further explored to find the best solutions regarding the supervision of the public sector, as highlighted by the researchers analyzed; specifically, the independence of the audit, the communication strategies, and the responsibility of civil servants.

The results obtained can be useful to professionals in the field of auditing in the public sector due to the analysis and synthesis carried out in this study. Managers of public institutions can use them as a landmark in future strategies for better management of public funds. Regulators can identify the aspects that should be modified in the legislation to improve the activity of public sector auditors. In response to the research question; from the analyzed articles, it emerged that the audit in the public sector contributes substantially to the reduction of fraud and corruption phenomena, so that any form of control of the institutions leads to a better management of public funds and to the increase of citizens' trust in them. Therefore, citizens are the main stakeholders impacted by how public money is spent.

The limits of the conducted study are due to the selected sample, the works being taken from a single database (WoS), which led to the limitation of identifying other published works; however, the research can be extended through a bibliometric analysis of the articles indexed in other databases' data. This paper can also be a premise for quantitative research that can include variables in an econometric model to highlight the factors that influence the quality of the audit in the public sector.

**Author Contributions:** Conceptualization, D.-S.B. and C.-D.H.; methodology, D.-S.B. and C.-D.H.; software, C.-D.H. validation, C.-D.H.; formal analysis, D.-S.B.; investigation, D.-S.B.; resources, D.-S.B.; data curation, D.-S.B.; writing—original draft preparation, D.-S.B. and C.-D.H.; writing—review and editing, D.-S.B. and C.-D.H.; visualization, D.-S.B.; supervision, C.-D.H.; project administration, D.-S.B.; funding acquisition, D.-S.B. and C.-D.H. All authors have read and agreed to the published version of the manuscript.

**Funding:** This research received no external funding.

**Institutional Review Board Statement:** Not applicable.

**Informed Consent Statement:** Not applicable.

**Data Availability Statement:** The data presented in this study are available upon request from the corresponding author.

**Conflicts of Interest:** The authors declare no conflicts of interest.

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
