# Peer review of "Bibliometric Framing of Research Trends Regarding Public Sector Auditing to Fight Corruption and Prevent Fraud"

_jrfm, doi:10.3390/jrfm17030094_

Round 1
Reviewer 1 Report
Comments and Suggestions for Authors
The authors have made a great effort to collect and analyse data. I suggest them to summarise results observed from different studies about the topic and conclude the paper with the observed results.
Author Response
Thank you for your observations and for the opportunity to improve our manuscript and perform new corrections that have added significant value to our research endeavour!
We are very grateful for investing your time to analyse the paper and make very useful, encouraging, and thoughtful comments and recommendations. We have carefully read and considered each evaluation and thus performed major improvements of our manuscript, as requested, marked with green colour into the document.
The authors have made a great effort to collect and analyse data. I suggest them to summarise results observed from different studies about the topic and conclude the paper with the observed results.
R. Thank you for your suggestions, we included the recommended aspects, please see lines 375-380.
Reviewer 2 Report
Comments and Suggestions for Authors
1. Although the problem was raised, the research questions that will be answered in the study were not presented. The problem needs to be clearly defined. Research methodology should be clearly defined. Research questions, and objectives should be clearly defined in the introduction section.
22. Although the topic in general is not new and the gap authors try to cover is not clear, the paper proposes good interpretations and methdology. Therefore, research field and idea is considered related to the topic of the journal and presents a comprehensive analysis of the previous research in the field. In order to combat corruption and prevent fraud, the paper identifies trends and approaches in the field of auditing in the public sector. It then uses bibliometric analysis to achieve its goals. The findings demonstrated a research interest in this area, with the trend becoming more noticeable since 2017. The performance audit and the fight against corruption were the primary subjects covered, and the most pertinent research was done on samples from Nordic European nations.
3. By combining data about public sector performance and how it is tracked through audits, the paper adds to the body of current research. Through the detection of fraud and the provision of recommendations aimed at improving the efficiency of this sector's operations, the research presents the external audit as a critical component in the fight against corruption in the public sector. The paper uses bibliometric analysis on a large number of published papers to introduce the role that audit plays in the public sector in the fight against corruption and fraud prevention.
4. Although the method used is clear, sequential, and smooth, there should consider the institutional issues and its relation to the public sector strategies. Notwithstanding these encouraging findings, investigation should identify the most effective solutions for public sector.
55. The conclusions make sense in light of the facts and arguments put out, and they answer the main issue. However, authors also need to take into account the target group and how to utilize the findings.
6. The references used are appropriate and up-to-date and cover most parts of the research
7. Make sure the format throughout the paper are consistent
8. Below you will find the comments arranged by section:
Title:
· It is better if you reduce the title length.
Abstract:
· Looks good.
Keywords:
· Looks good.
Introduction
· The problem needs to be clearly defined
· Research methodology should be clearly defined
· Research questions and objectives should be clearly defined in this section.
Literature review
· What is the gap you try to cover?
Materials and Methods
· The tool used to collect the required need more explanations.
· It is good if you draw a chart to clarify the methodology
Results
· Why the annual evolution of published articles in year 2021 is greater than the other years before and after?
· It is good if you analyze the publications based on the region or continent.
· In table 7: sometimes you use bold format. Please unify the format throughout the paper.
· The consistency and sequence in this section is well organized
Conclusion:
· What is the target group of the study? How they can benefit from the results?
Author Response
Thank you for your observations and for the opportunity to improve our manuscript and perform new corrections that have added significant value to our research endeavour!
We are very grateful for investing your time to analyse the paper and make very useful, encouraging, and thoughtful comments and recommendations. We have carefully read and considered each evaluation and thus performed major improvements of our manuscript, as requested, marked with green colour into the document.

Reviewer 3 Report
Comments and Suggestions for Authors
The study of corruption and fraudulent practices in the public sector organizations has always been a very significant topic of research. The current study aims to explore it via bibliometric analysis.
The following comments and suggestions may help authors to improve their manuscript.
-Although authors have justified the application of bibliometric analysis in Section 3, however, it should have been substantiated in the Introduction section, particularly when developing research objectives.
-The review of literature is fine, however, the range of arguments is fewer and lacking cohesiveness.
-In the Results section, authors have given many figures and exhibits, however, they sound more descriptive in tone. The in-depth reasoning should be explored too. Authors should give convincing arguments to back up their findings.
Comments on the Quality of English Language
Some minor issues detected
Author Response
Thank you for your observations and for the opportunity to improve our manuscript and perform new corrections that have added significant value to our research endeavour!
We are very grateful for investing your time to analyse the paper and make very useful, encouraging, and thoughtful comments and recommendations. We have carefully read and considered each evaluation and thus performed major improvements of our manuscript, as requested, marked with green colour into the document.
The study of corruption and fraudulent practices in the public sector organizations has always been a very significant topic of research. The current study aims to explore it via bibliometric analysis.
The following comments and suggestions may help authors to improve their manuscript.
-Although authors have justified the application of bibliometric analysis in Section 3, however, it should have been substantiated in the Introduction section, particularly when developing research objectives.
R. Thank you for your suggestions, we included the recommended aspects, please see lines lines 51-56.
-The review of literature is fine, however, the range of arguments is fewer and lacking cohesiveness.
R. We included the recommended aspects in lines 73-76 and 132-135.
-In the Results section, authors have given many figures and exhibits, however, they sound more descriptive in tone. The in-depth reasoning should be explored too. Authors should give convincing arguments to back up their findings.
R. We included the recommended aspects in lines 169-171, 177-179, 234-236, 280-283, 375-380 and 408-413.
Reviewer 4 Report
Comments and Suggestions for Authors
Dear authors, i had the opportunity to revise the paper. I have few suggestions:
- Why was WoS category "Political science" not included in the bibliometric analysis? I would argue that some papers on that topic could be relevant for the analysis
- Table 6: Please provide average and cumulative number of citations, by journal
Author Response
Thank you for your observations and for the opportunity to improve our manuscript and perform new corrections that have added significant value to our research endeavour!
We are very grateful for investing your time to analyse the paper and make very useful, encouraging, and thoughtful comments and recommendations. We have carefully read and considered each evaluation and thus performed major improvements of our manuscript, as requested, marked with green colour into the document.
Dear authors, i had the opportunity to revise the paper. I have few suggestions:
- Why was WoS category "Political science" not included in the bibliometric analysis? I would argue that some papers on that topic could be relevant for the analysis
R. Thank you for your suggestions, we included the recommended aspects please see the lines 149-150, 177-180.
- Table 6: Please provide average and cumulative number of citations, by journal
R. Thank you for your suggestions, but unfortunately we could not include the 2 columns in table 6 because in the initially downloaded database we did not have this information, and now if we download the data again, all the citations are up to date, but the reference date of the study ours is June 30, 2023 (line 143). To implement your suggestion, we should have collected the data manually, therefore we gave up on this aspect, but we added a sentence that refers to the performance of journal articles through the accumulated citations (lines 280-283). Anyway, in table 5 and table 7 are mentioned citations on the relevant authors and articles, we consider that we have captured the essence of the performance of the most relevant articles through the existing citations. Thank you for your understanding.